# Slope Runoff Process and Regulation Threshold under the Dual Effects of Rainfall and Vegetation in Loess Hilly and Gully Region

**Qiufen Zhang, Xizhi Lv \*, Yongxin Ni, Li Ma and Jianwei Wang**

Yellow River Institute of Hydraulic Research, Henan Key Laboratory of Yellow Basin Ecological Protection and Restoration, Zhengzhou 450003, China
\* Correspondence: nihulvxizhi@163.com

**Abstract:** The rainfall in loess hilly and gully area is concentrated, and mostly comes in the form of rainstorms. The runoff on the slope caused by rainstorms is the main cause of serious soil and water loss in the loess hilly area, and the grassland vegetation has a good inhibitory effect on the runoff on the slope. Therefore, it is of great significance to reveal the role of grassland vegetation in the process of runoff generation, and the mechanisms for controlling soil erosion in this area. In this study, typical grassland slopes in hilly and gully regions of the loess plateau were taken as research objects. Through artificial rainfall in the field, the response rules of the slope rainfall-runoff process to different grass coverage were explored. The results show that: (1) With the increase in rainfall intensity, the inhibitory effect of grassland vegetation on slope runoff decreased, which was mainly reflected in the gradual decrease in runoff rate and runoff coefficient, and the time required to reach stability gradually shortened. (2) Under 60 mm/h rainfall intensity, the sensitivity of runoff coefficient to 31.5% of grass cover change is the lowest, and the cost performance of grass cover with 55% coverage is the highest. (3) Grass coverage inhibited slope runoff by changing the hydraulic characteristics of the slope, but this effect was only obvious in low rainfall intensity and early rainfall. Rainfall in the loess hilly area is characterized by intense rain. The regulating effect of grass cover on slope runoff is not particularly significant under high intensity rainfall. If only considering the regulation of grassland vegetation on slopes, more than 60% grassland coverage is more efficient in inhibiting slope runoff under medium and low intensity rainfall.

**Keywords:** grass coverage; rainfall intensity; regulation threshold; slope runoff process; hydraulic parameter





## 1. Introduction

The ecological environment of the hilly and gully area of the Loess Plateau is fragile, and soil erosion is very serious. The source of serious soil erosion in this area is slope runoff erosion. Because it is located in a semi-arid zone where water resources are scarce, grassland is one of the main vegetation types in the loess hilly and gully region, which has a significant interference effect on slope runoff [1,2]. Clarifying the effect of grassland vegetation on slope runoff generation and its regulation mechanism in loess hilly and gully regions has important scientific significance for regional vegetation restoration and soil erosion control.

Previous studies have shown that plants can improve soil physical properties, such as increasing soil porosity and lifting soil hydraulic conductivity, etc., enhancing soil infiltration, and thereby promoting the slope runoff yield [3–6]. In addition, grassland vegetation has a retarding effect on runoff [7–9] and has an important influence on runoff characteristics and surface flow hydraulic parameters [10,11]. Specifically, grassland vegetation has a good effect on suppressing the generation and development of slope runoff, which is mainly because grassland vegetation can reduce raindrop energy and increase soil infiltration rate,

thereby reducing slope runoff, affecting slope runoff coefficient, and delaying the initial time of slope runoff [12–14]. Other scholars have studied the response relationship between the runoff process and hydraulic parameters of bare land, grassland, and shrubland. The results show that as the rainfall intensity increases, the difference in soil infiltration rate between the initial and final stages increases [15,16]. On the slope covered with grass vegetation, the slope flow pattern is transitional flow and turbulent flow; the Reynolds number and Froude number will be changed by the change of the grass vegetation cover; the retarding effect of overland flow due to the influence of grassland vegetation decreases with increased runoff [17–19].

The above-mentioned studies were conducted unilaterally from the perspective of hydraulics or hydrology. There are few studies that pay attention to the hydrodynamic characteristics of different grassland vegetation coverage under artificial rainfall. After decades of restoration, the vegetation area of the Loess Plateau has increased significantly, the runoff and sediment yield both decreased [20–22]. As the main vegetation type on the Loess Plateau, grassland still plays an important role in regulating runoff erosion. In the hilly and gully areas of the Loess Plateau, what are the reasons for the changes in the process of runoff generation on the slope under different grass coverage? What is the critical grass coverage threshold when the slope runoff process changes? These issues are not clear yet. Therefore, this research aims to reveal the response regularity of the rainfall-runoff generation process to different grass cover degrees, explore the regulation mechanism and mode of grassland vegetation on the slope runoff generation process, and quantitatively analyze the critical grass cover degree threshold under the change of the runoff generation process. The research results will further enrich the theory of regional runoff yield and confluence, provide support for the improvement of slope hydrological models and soil erosion models, and provide a basis for regional soil and water conservation and ecological construction.

## 2. Materials and Methods

### 2.1. Site Description

This study was conducted in Tianshui Soil and Water Conservation Test Station of the Yellow River Conservancy Commission in the Loess Hilly and Gully Region, which is located in Luoyugou watershed in Tianshui City, Gansu Province ($105°30'$–$105°45'$ E, $34°34'$–$34°40'$ N), as shown in Figure 1. The basin is narrow and long, similar in shape to a feather, with an area of 72.79 km$^2$. Luoyugou watershed is located in the continental monsoon climate zone, with drought in winter and spring, and concentrated precipitation in summer and autumn. The average annual rainfall is 531.1 mm, evaporation is 1293.3 mm, aridity index K = 1.30, and annual average temperature is 10.7 °C in this watershed. There are 1 microclimate observation station, 6 rainfall stations, and several field runoff observation plots are set up in this watershed. The soil type in the study area is mainly Malan loess [23]. The study area provides good field experimental conditions for the development of this study.

### 2.2. Experimental Design

In this study, the most common herb in the hilly and gully regions of the Loess Plateau, *Medicago sativa* L., was selected as the research object. Based on the principles of smooth slope surface, low human disturbance, nearby water sources and power sources, and no deep leakage, we set up a control experiment with six grass coverage levels on a natural slope of 15 degrees according to the point-and-frame method (Figure 2). These six vegetation coverage levels were two low-coverage grassland, 20% and 35%, respectively; two medium-coverage grasslands, 50% and 65%, respectively; one high-coverage grassland, 80%; and one bare slope set as a control group.

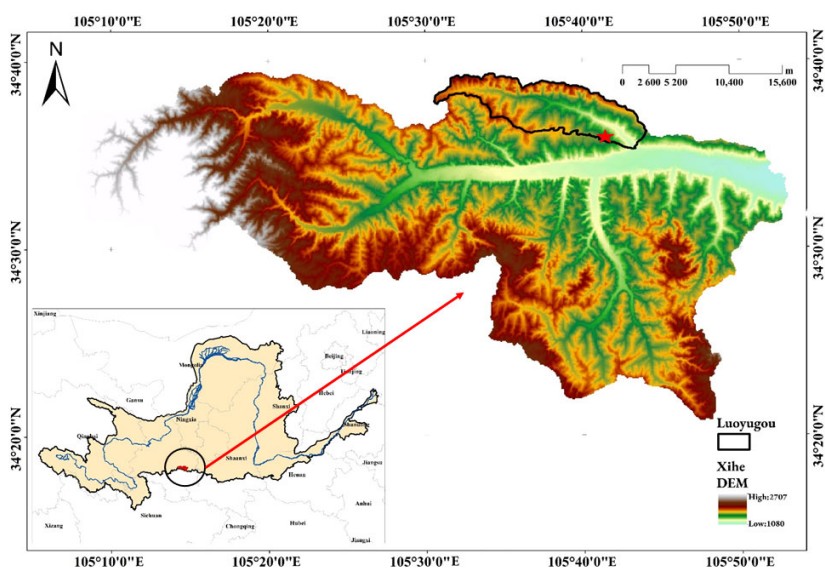

**Figure 1.** Geographical location of the study area.

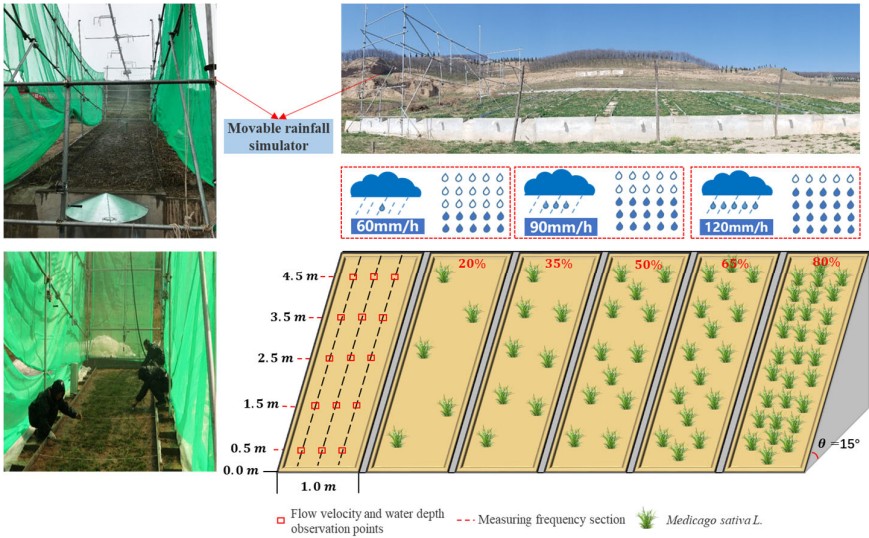

**Figure 2.** The schematic diagram of plot setting and monitoring points of slope experiment.

According to the average annual rainfall characteristics of the Loess Plateau, the annual maximum daily rainfall intensity in the study area ranges from 32.7 mm/d to 110.5 mm/d, and the average daily rainfall intensity of the study area is approximately 59.2 mm/d [24]. Thus, three different rainfall intensities (60 mm/h, 90 mm/h and 120 mm/h, each rainfall event lasting 60 min) were selected to study the basic runoff process under different rainfall characteristics in this study. During the 60 min period, if the slope runoff yield reaches a stable state, a stable runoff coefficient will be obtained, which is expressed as $R_{60}$ in this paper.

### 2.3. Experimental Measurement Methods

Before each rainfall experiment, a rain intensity of 30 mm/h was used to carry out the early rainfall until the slope runoff yield, and after the early rainfall, it was covered with plastic cloth and stood for 24 h to ensure that the soil moisture content in the early stage of each test was essentially consistent. Before and after each test, the soil bulk density was determined by the ring knife method on the slope, and the soil water content was determined by the drying method. The slope flow velocity was measured by the dyeing method, and five measurement sections were set, which were, respectively, located at

0.5 m, 1.5 m, 2.5 m, 3.5 m, 4.5 m from the bottom of the slope, and 3 points were taken on each measurement section for the measurement of flow velocity; the distance between the measuring point and the cell boundary was 0.3 m, 0.5 m, and 0.7 m, respectively. Each measurement point was measured 6 times and the average value was used, and the final average flow rate was obtained by correcting according to the flow pattern. The depth of rill flow was measured synchronously by digital stylus (accuracy 0.01 mm) at the flow rate measurement point, and the final value of each measurement point is the average of the four measurement results. After setting the rainfall intensity, we carried out the rainfall, recorded the time of rainfall, recorded the time of production flow when the slope flow was generated, and collected water samples. For the first 10 min, the sampling frequency was 1 time per minute, and was then adjusted to 1 time in 2 mins. The sampling of the depth of rill flow was carried out by special personnel at the same time.

The artificial rainfall simulation device in the field is a trough-type downward spray artificial rainfall simulation machine developed by Beijing Normal University and Beijing Jiaotong University. Rollers are installed on the base of the rainfall simulation machine to ensure that it can be temporarily moved above the runoff plot to be measured according to the experimental needs. The nozzle model of the rainfall machine is Spraying Systems Co. Veejet 80150, the water pressure is 0.04 MPa, and the nozzle height is above 2.5 m to ensure the end speed can be reached when the raindrops fall. There are 10 levels of rain intensity that can be simulated, ranging from 20 to 200 mm/h. The rain intensity can be adjusted by controlling the swing frequency of the nozzle.

### 2.4. Hydraulic Parameter and Soil Infiltration Rate Calculation

The hydraulic parameters of slope flow can be specifically expressed by flow velocity, Reynolds number, Froude number, and drag coefficient. The dimensionless parameter Reynolds number *Re* reflects the ratio of runoff inertial force and viscous force, which is an important parameter to determine the flow pattern of water flow. The Reynolds number is calculated according to Equations (1) and (2):

$$Re = \frac{vR}{v} \tag{1}$$

where $R$ is hydraulic radius (cm); $v$ is the flow velocity rate (cm/s); $v$ is the kinematic viscosity coefficient (cm$^2$/s). Among which,

$$v = \frac{0.01775}{(1 + 0.0337t + 0.000221t^2)} \tag{2}$$

where $t$ is the temperature of the water during the test.

The Froude number reflects the ratio of the inertial force of water flow to gravity. It is a dimensionless parameter and an important parameter to characterize the flow state of water flow. The formula for calculating Froude number is Equation (3):

$$Fr = \frac{v}{\sqrt{gh}} \tag{3}$$

where $g$ is the gravitational acceleration (m$^2$/s); $h$ is the depth of rill flow (m).

The drag coefficient $f$ and Manning roughness coefficient $n$ are calculated as Equations (4) and (5):

$$f = \frac{8gRJ}{v^2} \tag{4}$$

$$n = \frac{R^{\frac{2}{3}} J^{\frac{1}{2}}}{v} \tag{5}$$

where $J$ is the hydraulic slope for uniform flow, $J = \sin\theta$, $\theta$ is the slope of the plot, 15°.

The soil infiltration capacity is usually expressed by infiltration rate. Infiltration rate represents the amount of water that can be stored by soil in a certain area within a certain period of time. The formula is:

$$i = Pcosa - \frac{10V}{St} \tag{6}$$

where $i$ is the slope soil infiltration rate (mm/min); $P$ is rainfall intensity (mm/min); $a$ is slope degree (°); $V$ is the runoff generated during rainfall time (mL); $S$ is the vertical projection area of the plot (cm$^2$); $t$ is the rainfall time (min).Part of the experimental data (Table S1: The raw data) has been uploaded as Supplementary Materials.

## 3. Results

### 3.1. Runoff Yield Process in Different Rainfall and Vegetation Conditions

The curves of runoff–rainfall time under different rainfall intensities are shown in Figure 3. It can be seen that at the beginning of the experiment, about 15 min before, the runoff increased rapidly, and then increased slowly and gradually stabilized, which reflected that the soil infiltration was stabilizing after 15 min of rainfall, and was even close to saturation in bare slope. The time variation characteristics of runoff rate (Q, L/min) under different vegetation coverage are obviously different. The one way ANCOVA result with multiple sets of data showed that there was a significant difference ($p < 0.05$) between the bare slope and other grassland covered slopes, especially under 60 mm/h rainfall intensity, which indicated that the grassland coverage had a significant inhibitory effect on the slope runoff generation. As the grassland coverage increases, the stable runoff coefficient increased under any rainfall intensity. Especially under the rainfall intensity of 60 mm/h, the stable runoff rate was 0.04 L/min on the coverage of 80% slope and 9.30 L/min on the bare slope, which indicated that when the grassland vegetation coverage reached to 80%, it could minimize 95% of the runoff than bare slope. Under the rainfall conditions of 90 mm/h and 120 mm/h, the final stable runoff rate of the bare slope was 13.4 L/min and 19.1 L/min, respectively, and the final stable runoff rate of the 80% coverage slope was 3.7 L/min and 8.7 L/min accordingly, which can be reduced by 72.2% and 54.8% individually than bare slope.

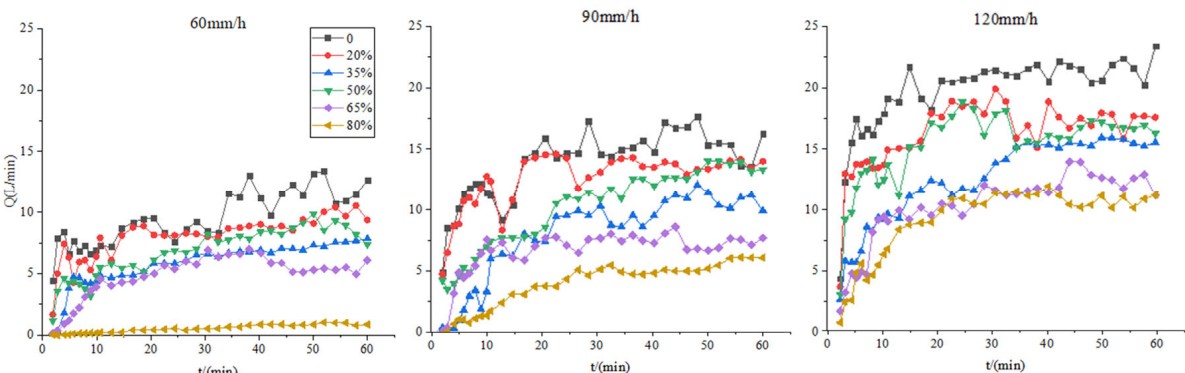

**Figure 3.** Runoff generation on slopes under different experimental conditions.

### 3.2. Hydraulic Parameters Change in Different Rainfall and Vegetation Conditions

#### 3.2.1. Flow Velocity

The temporal variation of overland flow velocity under different grassland vegetation coverage of the three rainfall intensities was shown in Figure 4. It can be seen from the figure that the slope flow velocity generally shows an increasing trend with the duration of rainfall and tends to be stable after a period of time, but the rainfall duration when flow rate reaches stability is different under different treatments. Under the rainfall intensity of 60 mm/h and 90 mm/h, the velocity of overland flow reached a stable value in about 25~30 min, while under the rainfall intensity of 120 mm/h, the slope flow velocity reached a stable value at about 20 min. This result indicated that with the increase in rainfall intensity,

the time required for slope flow velocity to stabilize was shorter. Under the same vegetation cover, the greater the rainfall intensity, the greater the slope runoff velocity.

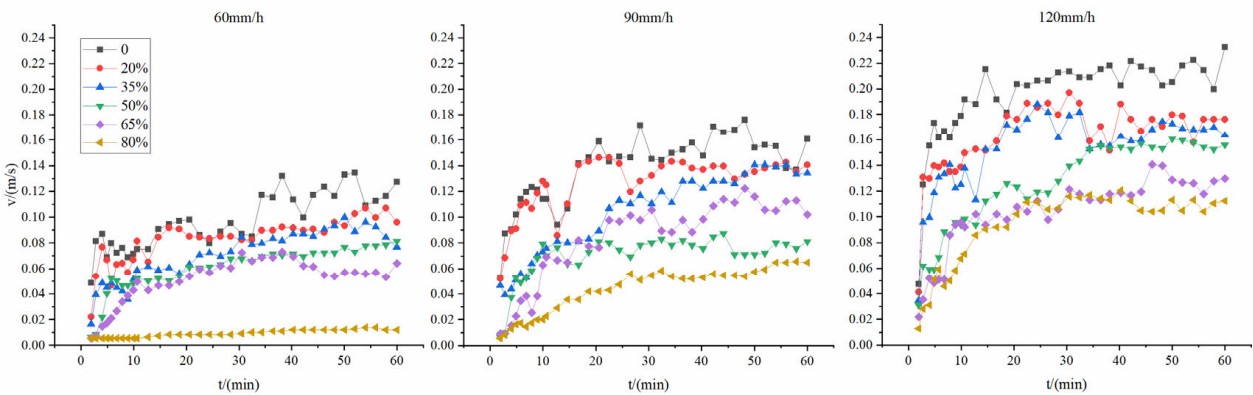

**Figure 4.** Flow velocity of overland flow under different rainfall intensities.

### 3.2.2. Reynolds Number

The $Re$ reflects the ratio between inertial force and viscous force; whether its value exceeds 580 is a dimensionless number to determine whether the flow pattern is laminar or turbulent. It can be seen from Figure 5 that, under different rainfall intensities, the $Re$ changes of each grassland coverage showed an increasing trend during the experiment. This growth trend was not particularly intense under the rainfall intensity of 60 mm/h but was very obvious on the slopes of 90 mm/h and 120 mm/h rainfall intensity, though the growth process of $Re$ was different under different rainfall intensities.

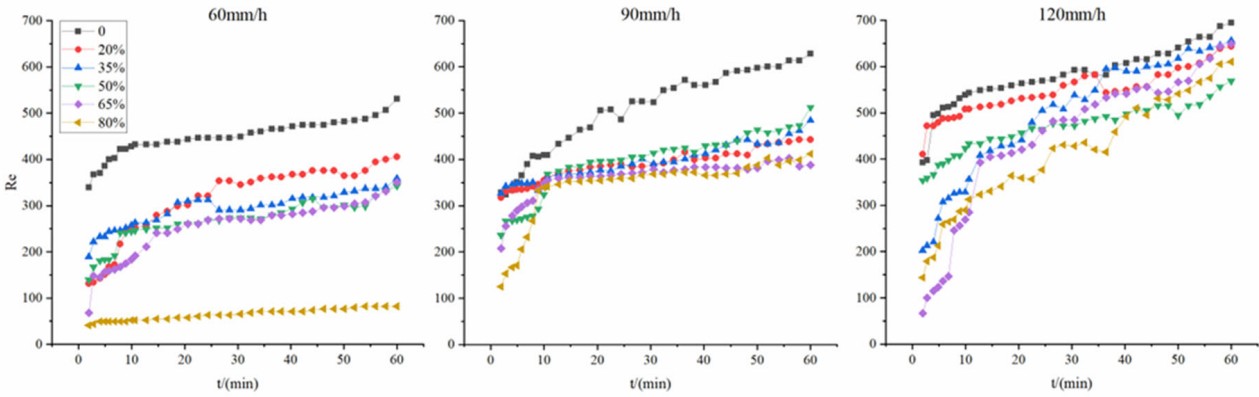

**Figure 5.** Variation of Reynolds Number under Different Test Conditions.

Under the rainfall intensity of 60 mm/h, $Re$ only increased rapidly in the first 15 min, and the growth trend tended to be stable after 25 min. Under the rainfall intensity of 90 mm/h, $Re$ increased particularly rapidly in the first 8 min, and slowly increased and gradually stabilized after 15 min. Under the rainfall intensity of 120 mm/h, $Re$ increased particularly rapidly in the first 15 min, then the growth slowed down but continued, and at the end of the experiment, the growth of $Re$ still showed signs of continuing.

It is worth mentioning that when the slope is bare, the $Re$ generally showed a trend of continuous growth at any rainfall intensity. However, under the conditions of 90 mm/h and 120 mm/h rainfall intensity, the $Re$ exceeded 580 in 43 min and 27 min, respectively, indicating that the slope flow changes from laminar to turbulent. Compared with the curve with vegetation coverage, it can be seen that vegetation coverage could significantly reduce $Re$, especially when the rainfall intensity was 60 mm/h and 90 mm/h. When the vegetation coverage reached 80%, the $Re$ at 60 m/h rainfall intensity was maintained at an extremely low level, indicating that the flow pattern was stable.

### 3.2.3. Froude Number

The *Fr* reflects the relationship between the inertial force of the water flow and gravity and can distinguish the flow pattern of the water flow. It is the basis for analyzing whether the slope water flow is rapid or slow. Sediment kinematics research shows that when the *Fr* value is less than 1, the water flow is slow; when the *Fr* value is greater than 1, the water flow is rapid.

As shown in Figure 6, under the conditions of different rainfall intensities and different grassland vegetation coverages, *Fr* showed a gradual downward trend as the rainfall continued. Under the same rainfall intensity, the *Fr* of different grassland vegetation coverage was significantly different. Under the same coverage, the difference of *Fr* between different rainfall intensities was smaller than that of the same rainfall intensity, indicating that the influence of rainfall intensity on *Fr* was more significant.

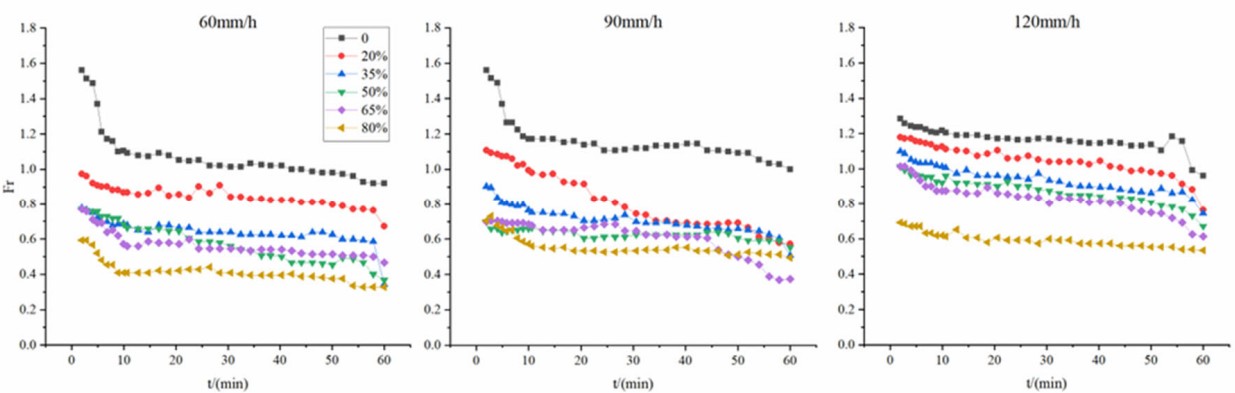

**Figure 6.** Variation of Froude number under different test conditions.

In this experiment, with continuous rainfall, for the slope under the rainfall intensity of 60 mm/h, the *Fr* of each coverage slope showed a rapid downward trend in the first 10 min, and then stabilized, especially on the bare slope. Under the rainfall intensity of 90 mm/h, the decreasing trend of the bare slope slowed down in the first 10 min, while other slopes did not change significantly. When the rainfall intensity reached 120 mm/h, the change trend of *Fr* in each coverage was gentle and the difference was small.

### 3.2.4. Drag Coefficient

It can be seen from Figure 7 that, under different rainfall intensities, the slope resistance coefficient of each coverage showed an upward trend with the increase in rainfall duration. The higher the vegetation coverage, the higher the average level of resistance coefficient.

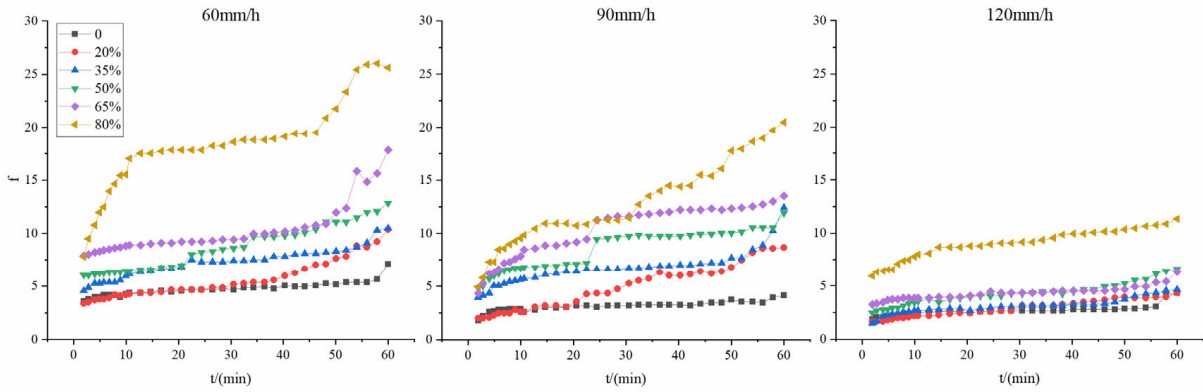

**Figure 7.** Variation of resistance coefficient under different test conditions.

Under the rainfall intensity of 60 mm/h, the resistance coefficient of the bare slope changed significantly, especially in the first 10 min, when the resistance coefficient increased

rapidly, and then tended to be gentle, while in other slopes with different grassland vegetation coverage, the resistance coefficient increased more slowly, and the change was not significant with continued rainfall. Under the rainfall intensity of 90 mm/h, the variation regularity of resistance coefficient was close to that of 60 mm/h rainfall intensity. In addition to the significant change trend of the bare slope resistance coefficient, the resistance coefficient of other slopes increased slowly. Under the rainfall intensity of 120 mm/h, the change of the resistance coefficient of each coverage was not significant, and the resistance coefficient was relatively small. Though the resistance coefficient of the bare slope was greater than 5, the resistance coefficient of other slopes was less than 5. From the overall trend, the resistance coefficient increased relatively little in the early stage of rainfall, but the increase in the resistance coefficient was relatively bigger in the later stage. The smaller the rainfall intensity, the greater the resistance coefficient. High vegetation coverage can significantly increase the resistance coefficient of overland flow.

### 3.3. Response Threshold of Slope Runoff Characteristics to Grassland Cover Changes

#### 3.3.1. Response Threshold of Runoff to Vegetation Coverage Change

Figure 8 shows the response of stable runoff coefficient ($R_{60}$) to vegetation coverage change. When the grassland coverage changed from 0 to 80%, the $R_{60}$ decreased, regardless of the rainfall intensity. In general, the descending order of different rainfall intensities was 120 mm/h > 90 mm/h > 60 mm/h, but the response process was different. When the coverage was less than 10%, the variation trend of $R_{60}$ between different rainfall intensities was roughly the same, and the difference was not obvious. When the coverage exceeded about 10%, the difference changed significantly. Especially under the rainfall intensity of 60 mm/h, the response extent of $R_{60}$ increased first, then decreased and finally increased again with the vegetation coverage increase. After the derivative calculation of the fitting function under 60 mm/h rainfall intensity, we found that the inflection point appeared at about 31.5%. When the vegetation coverage reached about 55%, the difference of $R_{60}$ under different rainfall intensities was the largest, indicating that there might be a sensitivity threshold between $R_{60}$ and grassland coverage. Under low rainfall intensities, the vegetation with coverage of 55% has the highest cost performance in regulating runoff. Under the condition of high intensity rainfall, the sensitivity between $R_{60}$ and grassland coverage was relatively stable, and the advantage of vegetation coverage was not obvious.

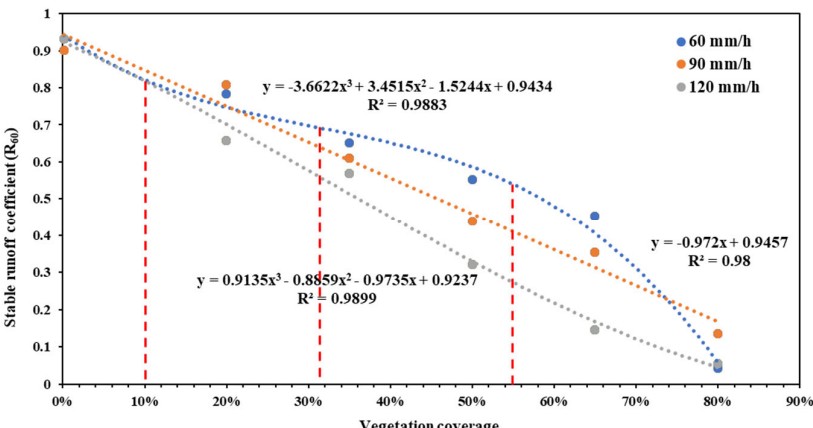

**Figure 8.** The variation trend of stable runoff coefficient ($R_{60}$) with the increase in vegetation coverage.

#### 3.3.2. Response Threshold of Hydraulic Parameters to Changes in Grassland Coverage

The analysis of hydrodynamic parameters variation with vegetation coverage change under different rainfall intensities was shown in Figure 9. Among them, *v*, *Re*, and *Fr* all showed a decreasing trend as the vegetation coverage increased. At the same time, the greater the rainfall intensity, the greater the values of *v*, *Re* and *Fr* under the same vegetation coverage. On the contrary, the relationship between *f* and vegetation coverage and rainfall

intensity was the opposite. However, the response degree of hydraulic parameters to vegetation coverage was significantly different ($p < 0.05$). By calculating the ratio of the absolute value of the hydraulic parameter difference between the maximum and minimum vegetation coverage to the value in bare slope, we compared the influence of vegetation on hydraulic parameters under different rainfall intensities.

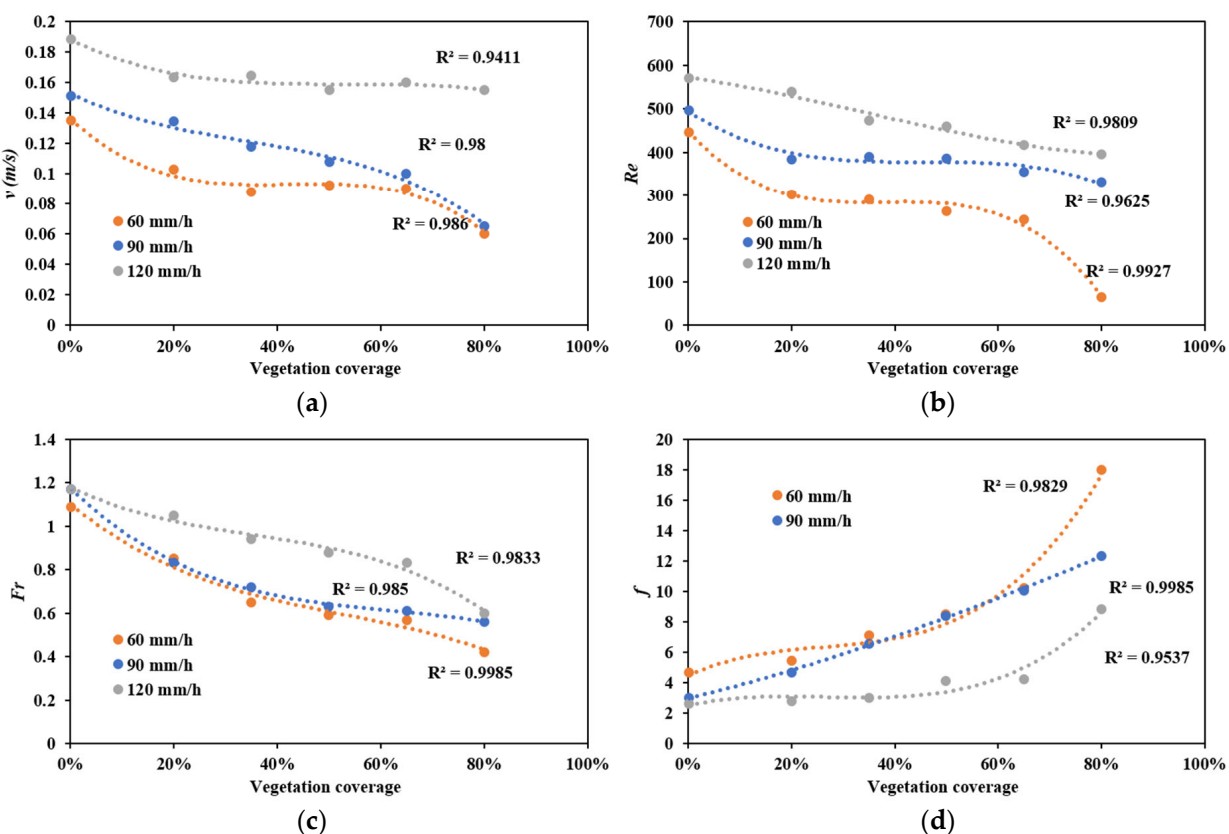

**Figure 9.** The variation trend of hydraulic parameters ((**a**): flow velocity; (**b**): *Re*; (**c**): *Fr*; (**d**): *f*) with the increase in vegetation coverage.

For flow velocity in Figure 9a, when the vegetation coverage changed from 0 to 80%, the *v* value under 120 mm/h rainfall intensity was the highest with the smallest variation amplitude, just 17.58%, which was smaller with 55.23% and 56.98% variation, respectively, under 60 mm/h and 90 mm/h rainfall intensity. The curve in 60 mm/h has a small stationary interval between 0.3~0.5, which further indicated that there was a low sensitivity threshold interval for flow velocity to changes of grassland coverage in low rainfall intensity. In contrast, the threshold effect was not obvious under 120 mm/h and 90 mm/h rainfall intensity.

For *Re* in Figure 9b, when the vegetation coverage changed from 0 to 80%, the variation amplitude under 120 mm/h and 90 mm/h rainfall intensity was relatively small, just 30.70% and 33.35%, respectively, while the variation amplitude was higher under 60 mm/h rainfall intensity, with 85.57%. The curve in 60 mm/h has a small stationary interval between 0.3~0.5, which further indicated that there was a low sensitivity threshold interval for *Re* to changes of grassland coverage in low rainfall intensity. In contrast, the threshold effect was not obvious under 120 mm/h and 90 mm/h rainfall intensity.

For *Fr* in Figure 9c, the variation trend of Fr with vegetation coverage increase was the same under 60 mm/h and 90 mm/h rainfall intensity. When the vegetation coverage was between 0.2~0.6, *Fr* under 120 mm/h rainfall intensity was significantly higher than otherwise. There was no significant difference between bare land and slope with 80% coverage under different rainfall intensities.

For *f* in Figure 9d, *f* was positively correlated with vegetation coverage change generally. When the vegetation coverage changed from 0 to 80%, *f* value under 120 mm/h and 60 mm/h rainfall intensity increased slowly before 20% and relatively fast after it, while the change rate essentially remained stable under 90 mm/h. Under the same vegetation coverage, the greater the rainfall intensity, the smaller the *f* value of runoff, which means that the resistance of overland flow decreases with the increase in rainfall intensity.

## 4. Discussion

The regulation effect of grassland vegetation on runoff is mainly manifested in: (1) improving soil texture and increasing soil infiltration rate [25–27]; (2) reducing raindrop energy and intercepting rainwater [28]; (3) increased surface roughness and runoff resistance [29,30]. The runoff regulation ability of vegetation with different vegetation coverage is different [31]. In this study, the instantaneous runoff under various rainfall intensities showed a rapid increase trend in the first 10–15 min, mainly due to the interception and infiltration effects of grassland vegetation and soil on precipitation. As shown in Figure 10, by observing the time change process of soil infiltration, we found that, with the increase in rainfall time, the soil infiltration rate gradually decreased and tended to be stable after 15 min. This also means that the loss of the head of the initial rainfall was large, and it gradually decreased with the precipitation duration. According to the principle of water balance, the runoff rate gradually increases as the rainfall continues. After 15 min, as the water storage capacity of vegetation reached saturation and soil infiltration rate turned stable, the loss of precipitation decreased, and the increase trend of runoff tended to be flat, and, finally, showed a stable trend. Under the rainfall intensity of 120 mm/h, the effect of grassland vegetation on runoff was not significant. Even if the grassland coverage reaches 80%, the effect of reducing slope runoff was still not significant. This is because when the rainfall intensity is too high, the grass canopy interception and soil water tended to reach saturation quickly [32], the precipitation rate is also far greater than the infiltration rate, and thus most rainfall is converted into runoff. High rainfall intensity has a primal influence on infiltration excess runoff, which was also been confirmed in similar studies in arid areas [33,34]. Therefore, under extreme rainfall conditions, the regulation and control of grassland vegetation on slope runoff is limited.

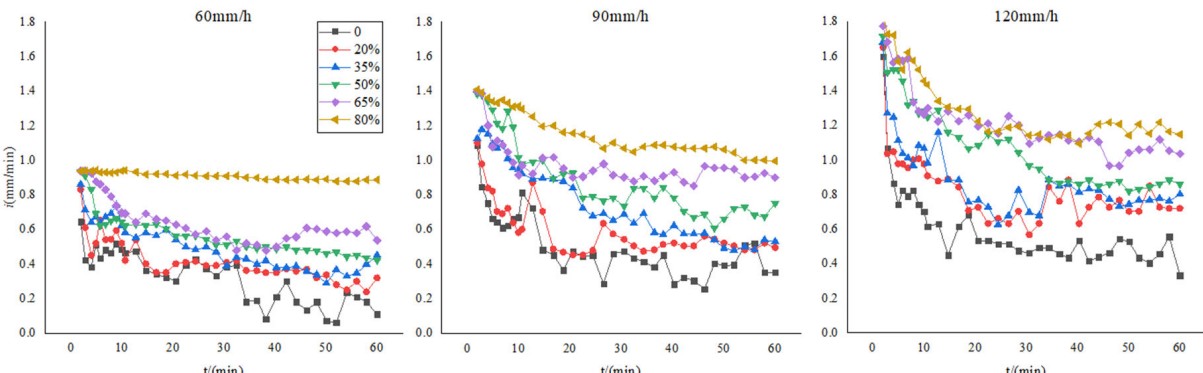

**Figure 10.** Variation of soil infiltration rate under different test conditions.

Regarding the regulation threshold of vegetation on runoff, a large number of previous studies have not determined a specific value. Research by scholars analyzing the effective coverage and critical coverage of grass cover indicated that when the grassland vegetation coverage is 60–80%, it can effectively prevent soil erosion [35]. Compared with bare slopes, grassland vegetation cover has a limiting effect on runoff of about 80% [18,36,37]. In this study, under the rainfall intensity of 60 mm/h, when the grassland vegetation coverage was greater than 60%, the $R_{60}$ presented a rapid decline trend, and this trend was not significant under the other two rainfall intensities, which was also due to the excessive rainfall intensity. As a result, grassland vegetation and soil are not fully functional in runoff

inhibition. Based on the 80% inhibition effect of grassland vegetation, under 60 mm/h and 90 mm/h rainfall intensity, the obtained coverage thresholds are 75% and 90%, respectively. A similar study suggested that vegetation cover persisted above 70% throughout the year, meeting the threshold level recommended to keep surface runoff and soil loss to minimum [38]. However, under the rainfall intensity of 120 mm/h, the regulation and control of grassland vegetation on runoff is not significant. When there is a heavy rainstorm, grassland vegetation can play a limited role in soil and water conservation. Therefore, in view of the rainfall in the loess hilly and gully area, a combination of forest, shrub, and grass should be adopted to control soil erosion in the area.

Regarding the influence of rainfall and underlying surface conditions on hydraulic parameters, the Reynolds number tends to increase with the increase in rainfall intensity, and rainfall intensity plays a leading role in the change of Reynolds number [39]. The rainfall intensity is negatively correlated to flow resistance coefficient, and grassland vegetation has a significant impact on the resistance coefficient [15,40,41]. In this experiment, the changes in hydraulic parameters were affected by the underlying surface conditions. The drag coefficient initially increased slowly, and then increased rapidly. The occurrence of runoff process was accompanied by the generation of rills on the slope. With the development of rills, the depth of rills increased, which in turn changed the roughness of the slope [42,43]. Under the experimental conditions, with the passage of rainfall events, the rill morphology also changed the flow pattern of the water flow, and the velocity of the slope flow also changed. Compared with the bare slope, the Froude number and Reynolds number of the grassland vegetation covered slope were significantly reduced. This is because the hindrance of the stem and root of the grassland vegetation increases the friction of the slope flow and the porosity of the soil, and increases the soil infiltration rate, thus affecting the hydraulic parameters. In addition, many researchers believe that slope degree will also affect the slope runoff capacity, infiltration characteristics, and hydraulic parameters [44,45]; that it can then affect the soil erosion of the slope [46]; and that the relationship between them will be different under different rainfall intensities [47,48]. However, due to the limitation of simulated rainfall conditions on natural slopes in the field, this paper has not considered the dual effects of vegetation and rainfall on runoff and sediment yield under different slope conditions. Relevant experiments are needed in the future to further improve the current theoretical results.

**5. Conclusions**

In this study, through the field simulation rainfall experiment, the slope runoff process under different rainfall intensity and vegetation coverage conditions was compared and analyzed, and the following conclusions were drawn: (1) With the increase in rainfall intensity, the inhibitory effect of grassland vegetation on slope runoff decreased, which was mainly reflected in the gradual decrease in runoff rate and runoff coefficient, and the time required to reach stability gradually shortened. When the rainfall intensity is 60 mm/h, 80% grassland coverage can reduce 95% of the slope runoff. When the rainfall intensity was 90 mm/h, the same grassland coverage could only reduce 72.2% of slope runoff. When the rainfall intensity was 120 mm/h, the same grassland coverage could only be reduced by 54.8% at the beginning of the rainfall event. (2) Under the condition of 60 mm/h rainfall intensity, the sensitivity of runoff to grass cover change has a threshold effect. When the coverage exceeds 10%, the sensitivity of runoff coefficient to grass cover change begins to decrease gradually. When the coverage is 50–60%, the sensitivity of runoff coefficient to grass cover change begins to change. With the increase in vegetation coverage, the sensitivity of runoff coefficient to grass cover change increases significantly. Under the condition of high intensity rainfall, with the increase in vegetation, the sensitivity of runoff to grass change tends to be consistent, and the threshold effect is not obvious. (3) The regulation of grass on runoff is mainly achieved by affecting the slope hydraulic parameters. The increase in grass coverage can effectively increase the surface roughness, reduce the overland flow velocity, and stabilize the overland flow pattern, but the effect

is obvious only in low rainfall intensity and early rainfall, and there are limitations under heavy rainfall conditions.

Therefore, in loess hilly and gully regions with frequent rainstorms, the vegetation restoration strategy should be formulated according to the characteristics of regional rainfall. In heavy rainfall areas, a vegetation restoration mode combining forest, shrub and grass should be adopted to make up for the deficiency of grass alone in reducing runoff and sediment.

**Supplementary Materials:** The following supporting information can be downloaded at: https://www.mdpi.com/article/10.3390/su15097582/s1, Table S1: The raw data.

**Author Contributions:** Q.Z. composed the paper. Q.Z. and Y.N. analyzed the data. Y.N., L.M. and J.W. assisted in the field simulated rainfall experiment. X.L. supervised this work. All authors have read and agreed to the published version of the manuscript.

**Funding:** This research was funded by the National Science Fund Project (Grant No. 52209022), the Excellent Youth Foundation of He'nan Scientific Committee (202300410541), the Youth Talent Support Program of Ministry of Water Resources in China, the National Science Fund Project (Grant No. U2243234, U2243601).

**Institutional Review Board Statement:** Not applicable.

**Informed Consent Statement:** Not applicable.

**Data Availability Statement:** The data presented in this study are available on request from the corresponding author.

**Conflicts of Interest:** The authors declare no conflict of interest.

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
