# Peer review of "Slope Runoff Process and Regulation Threshold under the Dual Effects of Rainfall and Vegetation in Loess Hilly and Gully Region"

_sustainability, doi:10.3390/su15097582_

Round 1

Reviewer 1 Report

This manuscript presents the difference in hydraulic parameters on a sloped surface based on different vegetation coverages. The study is well designed and presented. I recommend only minor changes, mostly clarifications, which are listed below in specific comments:

Lines 54-55: I think this sentence needs to be reworded to say “the retarding effect of overland flow due to the influence of grassland vegetation decreases with increased runoff.”

What is the grade of the slopes? Are they all the same?

Lines 94-97: I would suggest using a different word than gradient, which I initially confused with the grade of the slope rather than the vegetation density.

Figure 2 needs more description (i.e., Top-down view along slope) Describe difference between middle and right panels. 

Equations 1, 4, and 5 – what is “R”

Line 143: “The change regular between different coverage was different.” I do not understand this sentence.

I am confused by Figure 3. I would expect runoff to be highest over a bare slope and lowest over a vegetated slope, but the figure shows the opposite and the text indicates that 80% vegetation coverage minimizes runoff by up to 95%. Is this storage?

Lines 251-252: “…the runoff coefficient decreased slowly and then increased with the coverage increase…” I don’t think “increased” is the right word – is it supposed to describe the rate of change in the runoff coefficient? It seems like you are saying that the runoff coefficient increased with increased vegetation coverage.

Line 291: f looks like it is positively correlated with vegetation coverage change.

Line 341: The rainfall intensity is negatively correlated to what?

Author Response

Dear Editors and Reviewers:

Thank you very much for your second review of our paper of "Slope runoff process and regulation threshold under the dual effects of rainfall and vegetation in loess hilly and gully region" (sustainability-2229925). Those comments are all valuable and very helpful for revising and improving our paper, as well as the important guiding significance to our research. We have studied comments carefully and have made correction which we hope meet with approval. We have invited a native English scientist of relative research to polish and edit our wording and language, the details of the changes can be found in the manuscript. We hope the latest version of the manuscript could meet the journal’s standard. The main corrections in the paper and the responses to the reviewer’s comments are as flowing:

After carefully studying the comments and making corresponding correction, we have made corresponding changes. We hope the latest version of the manuscript could meet the journal’s standard. The main corrections in the paper and the responses to the reviewer’s comments are as flowing:

NOTE: All the Line numbers where revisions were made refer to the Manuscript with marked changes, the changes corresponding to the responses are highlighted in this revised version (#1Reviewer's response is marked in blue, #2Reviewer's response is marked in yellow, #3Reviewer's response is marked in green).

# Reviewer 1:

This manuscript presents the difference in hydraulic parameters on a sloped surface based on different vegetation coverages. The study is well designed and presented. I recommend only minor changes, mostly clarifications, which are listed below in specific comments:

Lines 54-55: I think this sentence needs to be reworded to say “the retarding effect of overland flow due to the influence of grassland vegetation decreases with increased runoff.”

Response:Thank you for your valuable comments. According to your suggestion, we have revised this sentence to "the retarding effect of overland flow due to the influence of grassland vegetation decreases with increased runoff" in line 64-55.

What is the grade of the slopes? Are they all the same?

Response:Thank you for your valuable comments. In order to increase the comparability of the results, we constructed a runoff test site on a uniform field slope with a slope of about 15 degrees. I am very sorry, we didn’t declare clearly in the experimental design section. According to your suggestion, we have added the description of slope information in the corresponding position (line 93-94) and reflected it in Figure 2.

Lines 94-97: I would suggest using a different word than gradient, which I initially confused with the grade of the slope rather than the vegetation density.

Response:Thank you for your valuable comments. I ' m sorry for the trouble caused by my improper wording. We have used different words to accurately express “grade of the slope” and “the vegetation density”, the specific modifications can be seen in line 94-97.

Figure 2 needs more description (i.e., Top-down view along slope) Describe difference between middle and right panels. 

Response:Thank you for your valuable comments. The first draft of Figure 2 does have some obvious problems: the details are not obvious and the expression is not clear enough. All reviewers have proposed modification suggestions to Figure 2. Therefore, we have redrawn this figure on the basis of the opinions of all reviewers, and specially added a section to introduce the implementation process of the experiment in detail. in “2.3. Experimental measurement methods”. We hope the revised figure and description can make reviewers to understand the layout of the experiment better. The specific modifications are in line 107-124.

Equations 1, 4, and 5 – what is “R”

Response: Thank you very much for pointing out my omission, the letter “R” represents “hydraulic radius (cm)” in this paper. According to your suggestion, we have described the physical quantity represented by the letter "R" below Equation (1) in line 142.

Line 143: “The change regular between different coverage was different.” I do not understand this sentence.

Response: I'm very sorry for the comprehension barrier caused by my poor writing, this sentence has been modified to "The time variation characteristics of runoff rate (Q, L/min) under different vegetation coverage are obviously different." in line 166-167.

I am confused by Figure 3. I would expect runoff to be highest over a bare slope and lowest over a vegetated slope, but the figure shows the opposite and the text indicates that 80% vegetation coverage minimizes runoff by up to 95%. Is this storage?

Response:Thank you for your valuable comments. You have pointed out a serious mistake in our drawing. Your suspicion is right. It is very ashamed that we have drawn the overall conclusion based on the original data. However, when drawing the line chart, due to carelessness, the order of the six broken lines sorted according to vegetation coverage is reversed, resulting in such a question. This is really a mistake that should not be, fortunately, you help us point it out. Benefit from your reminder, we have redrawn Figure3 in the original order.

Lines 251-252: “…the runoff coefficient decreased slowly and then increased with the coverage increase…” I don’t think “increased” is the right word – is it supposed to describe the rate of change in the runoff coefficient? It seems like you are saying that the runoff coefficient increased with increased vegetation coverage.

Response: I'm very sorry for the comprehension barrier caused by my poor writing, What I want to express is "the response extent of R60 increases first, then decreases and finally increased again with the vegetation coverage increase" according to the curve characteristics in Figure 8, I have revised it in line 275-276. In addition, we redefine the stable runoff coefficient of the ordinate axis to R60 in line 103-105 and use it here.

Line 291: f looks like it is positively correlated with vegetation coverage change.

Response: Thank you very much for pointing out my omission, I have checked and revised the sentence in line 320.

Line 341: The rainfall intensity is negatively correlated to what?

Response: Thank you very much for pointing out my omission, the original sentence is “The rainfall intensity is negatively correlated to flow resistance coefficient”, I have checked and revised it in line 378.

Author Response

Dear Editors and Reviewers:

Thank you very much for your second review of our paper of "Slope runoff process and regulation threshold under the dual effects of rainfall and vegetation in loess hilly and gully region" (sustainability-2229925). Those comments are all valuable and very helpful for revising and improving our paper, as well as the important guiding significance to our research. We have studied comments carefully and have made correction which we hope meet with approval. We have invited a native English scientist of relative research to polish and edit our wording and language, the details of the changes can be found in the manuscript. We hope the latest version of the manuscript could meet the journal’s standard. The main corrections in the paper and the responses to the reviewer’s comments are as flowing:

After carefully studying the comments and making corresponding correction, we have made corresponding changes. We hope the latest version of the manuscript could meet the journal’s standard. The main corrections in the paper and the responses to the reviewer’s comments are as flowing:

NOTE: All the Line numbers where revisions were made refer to the Manuscript with marked changes, the changes corresponding to the responses are highlighted in this revised version (#1Reviewer's response is marked in blue, #2Reviewer's response is marked in yellow, #3Reviewer's response is marked in green).

# Reviewer 2:

Dear Authors! I with interest read your manuscript entitled: “Slope Runoff Process and Regulation Threshold Under the Dual Effects of Rainfall and Vegetation in Loess Hilly and Gully Region”. The topic of article is relevant and fits to Sustainability journal scope, also its suitable for consideration in other MDPI journals, like: Land, Water, etc. The data and research of article is interesting, but unfortunately is not well written. I have some comments and suggestions:

  1. LL. 25-26. “Forest and shrub measures should be supplemented in order to control soil erosion more effectively”. You not conducted this research, so you could only suggest this statement.

Response:Thanks for your valuable comments. We have took your suggestion and removed this sentence from the abstract in line25.

  1. L. 42. Which soil parameters?

Response:Thanks for your valuable comments. To make the meaning clearer, we have further enumerated several important soil parameters and readjust the expression of the entire sentence in line 40-42.

  1. L. 81. “The aridity index K = 1.30”, seems too high for semi-arid area, which should be 0.20 < AI < 0.50 (https://en.wikipedia.org/wiki/Aridity_index).

Response:Thanks for your valuable comments. AI drought index is a world-recognized more commonly used drought index, the K drought index used in this paper, is a drought index calculated by Chinese scholars according to the spring precipitation and evaporation data of 140 weather stations in northwest China in 1971~2000, and has been confirmed to be more applicable to arid and semi-arid areas including the Loess Plateau of China, according to the division standard of K drought index, the drought index range of China's semi-arid and semi-humid areas is 1~1.49, and the interval in this paper meets the division standards..

  1. L. 84. “The soil type in the study area is mainly low-liquid-limit loess”. I never heard that such soil type could exist. Please give a soil type name according to WRB.

Response:Thanks for your valuable comments. After further review of data and literature, we determined that the main soil type in the study area was Malan loess, which is one of the Quaternary loess staging in China, and this soil naming method is more applicable in the Loess Plateau area. The specific modifications can be seen in line 84.

  1. L. 89-110. I not found any information about slope inclination. This factor very important for runoff formation. What about soil moisture conditions. Does soil was dry or wet? What about vegetation coverage range in the hilly and gully regions of the Loess Plateau? You could provide a map of vegetation density. Do you measure soil loss? In some cases the water runoff could be high but soil loss is low (when the turbidity of runoff is low).

Response:Thanks for your valuable comments. Here you actually asked four questions, which I will answer one by one.

  • Regarding the slope, thank you very much for pointing out our shortcomings, the runoff communities involved in the experiment are all 15 degrees, we have added relevant descriptions in the text (line93-94), and also marked and shown in the appropriate position in Figure 2;
  • Regarding the soil water content in the early stage, we have taken relevant measures to ensure that the soil moisture content of each runoff community is consistent before the simulated rainfall experiment. We added a description "Before each rainfall experiment, a rain intensity of 30mm/h was used to carry out the early rainfall until the slope runoff yield, and after the early rainfall, it was covered." with plastic cloth and stood for 24h to ensure that the soil moisture content in the early stage of each test was basically consistent.” in section 2.3 line 107-109.
  • Regarding the vegetation density map, in order to reveal the response threshold of runoff to vegetation cover, this experiment has set a gradient from 0~80% grass cover, which is enough to cover the range of grassland vegetation coverage on the Loess Plateau. In addition, the remote sensing imagery that typically characterizes vegetation cover is NDVI, which does not match the coverage used herein, and there does not appear to be any remote sensing imagery specifically characterizing vegetation density. Therefore, we do not seem to have the need to add vegetation cover to the map, so we have not made corresponding changes, and we hope that you will understand.
  • Regarding soil erosion, indeed, we have observed the amount of soil erosion at the same time in the experiment. However, since this paper focuses on the influence and threshold of hydraulic parameters on runoff changes under different vegetation cover, considering the limitations of the article and the structural integrity of the story, we have not mentioned the conclusions about soil erosion and its relationship with runoff, and plan to continue in another article.
  1. L. 100. “The average daily rainfall intensity of the study area is about 59.2 mm/d.” How many rainy days are occurring in the study area?

Response:Thank you for your valuable comments. According to the hydrological yearbook, the annual rainfall days in this region vary greatly, especially with climate change, short-duration heavy rainfall is frequent, and mostly concentrated in summer and autumn, the average daily rainfall described in this paper is the average amount in the past 60 years, which is only a general reference for the selection of rainfall intensity in this paper. As for “How many rainy days are occurring in the study area”, I am very sorry, it is really difficult to answer with an accurate number.

  1. L. 106. Figure 2. is not a rainfall simulator scheme, is a scheme of research. You have 6 variants, but on scheme only 3.

Response:Thank you for your valuable comments. The first draft of Figure 2 does have some obvious problems: the details are not obvious and the expression is not clear enough. All reviewers have proposed modification suggestions to Figure 2. Therefore, we have redrawn this figure on the basis of the opinions of all reviewers. We hope the revised figure can make reviewers to understand the layout of the experiment better. The specific modifications are in line107-124 and Figure 2.

  1. L. 140. What the reason of high runoff in first 15 min? Please give an explanation in Discussion section.

Response:Thank you for your valuable comments. We have added some new data about soil infiltration and expressed it in Figure 10. Combined with the water balance theory and Figure 10, we have explained it, the specific revision is in the line 337-345.

  1. Figure 3. Why runoff volume on bare soil is lowest? So no need to provide an anti-erosion practices.

Response:Thank you for your valuable comments. You pointed out a serious mistake in our drawing. Your suspicion is right. It is very ashamed that we have drawn the overall conclusion based on the original data. However, when drawing the line chart, due to carelessness, the order of the six broken lines sorted according to vegetation coverage is reversed, resulting in such a question. This is really a mistake that should not be, fortunately, you help us point it out. Benefit from your reminder, we have redrawn Figure 3 in the original order.

  1. L. 153. The units of runoff could be extra presented also in mm per ha and time (min or h, day). Your result (L/min) is runoff from your plot with area 5 m2, which is difficult to interpreting on big scale.

Response:Thank you for your valuable comments. We have carefully considered your suggestion. We understand that you want to convert the data of the cell to a larger scale to make the results in different scales more comparable. However, compared with the simple and homogeneous slope experimental conditions, the rainfall runoff process in large-scale space is more complex, and the matching rainfall spatial and temporal scales are also larger. Due to the existence of scale effect, the runoff yield per unit area and per unit time should be far less than the results of the plot, and the comparability between the two is not high. Finally, considering that this is a simulated rainfall experiment, we pay more attention to recording the data of a real experimental process and comparing it with the same scale research. Therefore, we did not modify the unit.

  1. Figure 8. The vegetation coverage better present in % (not as coefficient).

Response:Thank you for your valuable comments. We have took your suggestion and replaced the representation of vegetation cover with % instead of a coefficient in Figure 8 and Figure 9.

  1. Please prepare a Reference list according to journal rules.

Response:Thank you for your valuable comments. We have modified the bibliolist format in accordance with journal rules.

  1. I suggest to slight increase a number of References, as well compare your result with similar studies conducted in other countries (for example:

1) Evaluation of rainfall interception by vegetation using a rainfall simulator                                            

2) Effect of rainfall intensity and slope steepness on the development of soil erosion in the Southern Cis-Ural region (A model experiment)

3) Infiltration-excess runoff properties of dryland floodplain soil types under simulated rainfall conditions

 4) The use of rainfall simulation tests to assess the influence of vegetation density on soil loss on degraded rangelands in the Baringo District, Kenya

5) etc). Extra comments. I suggest you invite a English native colleague (specialist in soil erosion) and ask him to check your paper or use a English editing service. Your article is required of some corrections in terminology and text editing.

Response:Thank you for your valuable comments and thanks for your kind recommendation of these excellent articles for us. Based on your suggestions, we have added references to the discussion section and compared the results of similar studies with ours. The added references include 3 references you recommended and another one optional article. The specific modifications are in line 348-352 and line 367-369. In addition, I 'm sorry for the trouble caused by my poor English writing, we have invited a professional foreign language teacher to help us polish the text, hoping that the revised text will be smoother and easier to be understand.

Reviewer 3 Report

A manuscript Slope runoff process and regulation threshold under the dual effects of rainfall and vegetation in Loess Hilly and Gully region deal with a water runoff and several hydraulic parameters of water runoff under the different percentage of grass cover and a rainfall intensity. The manuscript has a standard structure, and can be interesting for the readers of Sustainability. However, it has several weaknesses in the current form. Therefore, following points should be consider before the publishing:

·       the presented study area is not relevant for the manuscript. The manuscript present results of experiments which can be relevant for any area of interest and can be simulated in the laboratory conditions.

·       The subchapter 2.2 Experimental design must be improved. The description of design does not correspond to presented results.

·       The reference for the Figure 2 (line 106) is not on the correct place. The figure does not show the rainfall device, but the experimental field.

·       Figure 2: the legend for the third pattern is missing. Please add it.

·       The meaning of R in equation 1, and following equations 4 and 5 is missing.

·       The information about determination of R is missing.

·       How the depth of rill flow was determined?

·       Authors are writing about the infiltration rate into soil, but no information about the soil is mentioned, including its characteristics and measurements.

·       In general, information about the measurement instruments is missing.

·       The description of results for different slope of the plot are not able to confirm – any graph or table should be added to see that.

·        Lines 163-164: “But the time…” the sentence is not understandable.

·       Figure 8: Stable runoff coefficient has not been mentioned earlier, and any information how it was determined is missing.

·       Figure 9: The title of the figure is not correct.

·       Due to submitting the manuscript into the international journal, it would be worth to add more foreign literature, especially into the discussion where the obtained results should be discussed with results of other researchers.

Author Response

Dear Editors and Reviewers:

Thank you very much for your second review of our paper of "Slope runoff process and regulation threshold under the dual effects of rainfall and vegetation in loess hilly and gully region" (sustainability-2229925). Those comments are all valuable and very helpful for revising and improving our paper, as well as the important guiding significance to our research. We have studied comments carefully and have made correction which we hope meet with approval. We have invited a native English scientist of relative research to polish and edit our wording and language, the details of the changes can be found in the manuscript. We hope the latest version of the manuscript could meet the journal’s standard. The main corrections in the paper and the responses to the reviewer’s comments are as flowing:

After carefully studying the comments and making corresponding correction, we have made corresponding changes. We hope the latest version of the manuscript could meet the journal’s standard. The main corrections in the paper and the responses to the reviewer’s comments are as flowing:

NOTE: All the Line numbers where revisions were made refer to the Manuscript with marked changes, the changes corresponding to the responses are highlighted in this revised version (#1Reviewer's response is marked in blue, #2Reviewer's response is marked in yellow, #3Reviewer's response is marked in green).

# Reviewer 3:

A manuscript Slope runoff process and regulation threshold under the dual effects of rainfall and vegetation in Loess Hilly and Gully region deal with a water runoff and several hydraulic parameters of water runoff under the different percentage of grass cover and a rainfall intensity. The manuscript has a standard structure, and can be interesting for the readers of Sustainability. However, it has several weaknesses in the current form. Therefore, following points should be consider before the publishing:

the presented study area is not relevant for the manuscript. The manuscript present results of experiments which can be relevant for any area of interest and can be simulated in the laboratory conditions.

The subchapter 2.2 Experimental design must be improved. The description of design does not correspond to presented results.

Response:Thank you for your valuable comments. Based on your comments and the comments of other reviewers, we have made significant changes in "2. Materials and Methods", add a section "2.3. Experimental measurement methods" to describe the details of the experiment more specifically, and modified "Figure 2" to more intuitively reflect the real scene of the experiment. I believe it will make you more clearly about the experiment.

The reference for the Figure 2 (line 106) is not on the correct place. The figure does not show the rainfall device, but the experimental field.

Response: Thanks for the suggestion, we have changed the position of the reference for the Figure 2 after the sentence "……we set up a control experiment with six grass coverage levels on a natural slope of 15 degrees according to the point-and-frame method " in line 94 based on your suggestion, where may be more appropriate we think. In addition, we have redrawn Figure 2 on the basis of the opinions of all reviewers, and the experimental scene was shown in detail.

Figure 2: the legend for the third pattern is missing. Please add it.

Response:Thank you for your valuable comments. The first draft of Figure 2 does have some obvious problems: the details are not obvious and the expression is not clear enough. All reviewers have proposed some modification suggestions to Figure 2. Therefore, we have redrawn this figure on the basis of the opinions of all reviewers. We hope the revised figure can make reviewers to understand the layout of the experiment better. The specific modifications are in line 125-126.

The meaning of R in equation 1, and following equations 4 and 5 is missing.

Response: Thank you very much for pointing out my omission, the letter “R” represents “hydraulic radius (cm)” in this paper. According to your suggestion, we have described the physical quantity represented by the letter "R" below Equation (1) in line 142.

The information about determination of R is missing.

Response: Thank you for your valuable comments. This question is similar to the previous question, we have described the physical quantity represented by the letter "R" below Equation (1) in line 142. Thank you for your careful review.

 How the depth of rill flow was determined?

Response: Thank you for your valuable comments. The depth of rill flow is measured synchronously by digital stylus (accuracy 0.01mm) at the flow rate measurement point, and the final value of each measurement point is the average of the four measurement results. We have supplemented the relevant descriptive text in the Research Methods section in line 118-124.

Authors are writing about the infiltration rate into soil, but no information about the soil is mentioned, including its characteristics and measurements.

Response: Thank you for your valuable comments. We have also observed soil moisture in the experiment. However, because the main research objects of this article are slope hydraulic parameters and vegetation coverage, considering the theme, structure and length of the article, we did not report the soil data separately in the article at the beginning. After seriously considering your suggestion, we found that it is necessary to add soil infiltration data, so we add the calculation method of soil infiltration in part 2.4 in line 153-159; in the discussion section, the corresponding figure (Figure 10) and text (line337-339) were added. We believe that the data obtained from the experiment will better support the current research conclusions compared with literature references.

In general, information about the measurement instruments is missing.

Response:Thank you for your valuable comments. Based on your comments, we have made significant changes in "2. Materials and Methods", add a section "2.3. Experimental measurement methods" to describe the details of the experiment more specifically. Our main device information has been mentioned in line 127-135, in addition, we have added a description about methods and equipment for monitoring slope runoff depth in line 118-120. As for the other equipment involved, most of them are more conventional, so we didn’t listed in detail due to the limitations of the article layout.

The description of results for different slope of the plot are not able to confirm – any graph or table should be added to see that.

Response:Thank you for your valuable comments. I 'm not sure if I understand your meaning correctly, but I suspect that what you say “The description of results for different slope of the plot are not able to confirm ”may refer to the unclear expression of the threshold and percentage values described in my results. Because my results are mainly based on trend analysis and quantitative analysis, the trend analysis is mainly expressed by line chart and curve chart, and the data used in the drawing is also based on the calculation method mentioned in the Materials and Methods section. The conclusion should be obvious and no controversy. Therefore, I guess your question comes from the fact that there is no clear basis for the quantitative results in the text. My specific reply is as follows:

  • The quantitative description of threshold mainly appears is in 3.3.1, of which 10 % and 55 % in 3.3.1 are estimated values, which can be seen intuitively on the figure. 31.5 % is the R60 change threshold as coverage increase under 60mm/h rainfall intensity, which is obtained by the derivative of the curve fitting equation. We have add the fitting equation of all curves to Figure 8 and the method expression “the response extent of R60 increases first, then decreases and finally increased again with the vegetation coverage increase. After the derivative calculation of the fitting function under 60mm / h rainfall intensity, we found that the inflection point appeared at about 31.5 %” in Line 275-278, which is convenient for readers to check.
  • The percentage values used to compare the influence of vegetation on hydraulic parameters under different rainfall intensities are in 3.3.2, which is confirmed by calculating the ratio of the absolute value of the hydraulic parameter difference between the maximum and minimum vegetation coverage to the value in bare slope. The related description have been added in Line 295-298. The original data will be uploaded as an attachment to facilitate the reader to test the results.

 Lines 163-164: “But the time…” the sentence is not understandable.

Response: I'm very sorry for the comprehension barrier caused by my poor writing, this sentence has been modified to "but the rainfall duration when flow rate reaches stability is different under different treatments." in line 187-188, we hope the revised sentences will improve your reading experience.

Figure 8: Stable runoff coefficient has not been mentioned earlier, and any information how it was determined is missing.

Response: Thank you very much for pointing out my omission, we have added a sentence “During the 60-minute period, if the slope runoff yield reaches a stable state, a stable runoff coefficient will be obtained, which is expressed as R60” in line 103-105 to define R60 and re-noted it in line247.

 Figure 9: The title of the figure is not correct.

Response:Thank you for your valuable comments. According to your suggestion, we have modified the title of Figure 8 to “The variation trend of stable runoff coefficient (R60) with the increase of vegetation coverage.” in line 268-269.

Due to submitting the manuscript into the international journal, it would be worth to add more foreign literature, especially into the discussion where the obtained results should be discussed with results of other researchers.

Response:Thank you for your valuable comments. Based on your suggestions, we have added some foreign literature in the discussion section and compared the results of similar studies with ours. The specific modifications are in line 348-352 and line 367-369.

Round 2

Reviewer 2 Report

Dear Authors! Thanks for revision, now the paper looks much better; I appreciate your efforts in improving of manuscript.

1.L. 94. 15% of slope. How many areas with similar slopes in study region/China? I suggest add a small discussion about impact of slope on runoff. (Some literature which probably will useful

https://doi.org/10.1016/j.catena.2019.02.004

https://doi.org/10.1007/s00477-014-0896-1

https://doi.org/10.1016/0341-8162(95)00003-B

https://doi.org/10.1134/S106422931709006X

https://doi.org/10.1016/j.geomorph.2010.12.004)

2. L. 110. You measured soil bulk density before and after experiment, but the results not presented and discussed in the text. And what the reason of it measuring?

3. L. 154. During infiltration water not “absorbed” by soil; infiltration is simply process of the downward entry of water into the soil through pores.

4. L. 378. What a value of correlation?

With best regards, Reviewer. Please keep going your research in soil erosion simulation, probably study with different slopes angle will have a interest for you. Good luck in future research!

Author Response

Dear Editors and Reviewers:

Thank you very much for your second review of our paper of "Slope runoff process and regulation threshold under the dual effects of rainfall and vegetation in loess hilly and gully region" (sustainability-2229925). Those comments are all valuable and very helpful for revising and improving our paper, as well as the important guiding significance to our future researches. According to the recommendations, we have made new changes and replies, hoping to meet the requirements of journals, editors and reviewers. Thank you very much for the valuable time you spent for my article.

NOTE: All the Line numbers where revisions were made refer to the Manuscript with marked changes, the changes corresponding to the responses are highlighted in yellow.

1.L. 94. 15% of slope. How many areas with similar slopes in study region/China? I suggest add a small discussion about impact of slope on runoff. (Some literature which probably will useful

https://doi.org/10.1016/j.catena.2019.02.004

https://doi.org/10.1007/s00477-014-0896-1

https://doi.org/10.1016/0341-8162(95)00003-B

https://doi.org/10.1134/S106422931709006X

https://doi.org/10.1016/j.geomorph.2010.12.004)

Response:Thank you very much for your valuable suggestion. From your literature and suggestions, we can see that you must have done a lot of research work on slope soil erosion, but there are some things I have to explain to you. Because what we carried out is simulated rainfall experiment in field, the simulated rainfall device we use is very heavy, once installed, it is very difficult to move ; in addition, the observation point is far away from the water source. The water we used in our experiment is extracted by a pump and stored in a water cellar, so the available water is very limited and very precious. Therefore, we do not give priority to the change of slope when choosing the slope, but consider the convenience of installation and movement of experimental equipment to ensure the normal operation of the experiment. I think you should also understand the difficulty of field experiments, so I express my gratitude and deep apology to you here.

In addition, the literatures you provided are very useful and have a good guiding role for our future research. We adopt all of them, and the specific modifications are in line 392-399.

  1. L. 110. You measured soil bulk density before and after experiment, but the results not presented and discussed in the text. And what the reason of it measuring?

Response:Thank you very much for your valuable suggestion. The measurement of soil bulk density is a routine operation to confirm whether there is a significant difference in the initial physical properties of the soil on the slope to be measured. Of course, it is generally not, because these slopes are close to each other and the spatial heterogeneity is relatively small. This operation is to make the experimental process more standardized, which is not directly related to the data of the text. I hope I have explained this problem clearly to you.

  1. L. 154. During infiltration water not “absorbed” by soil; infiltration is simply process of the downward entry of water into the soil through pores.

Response:Thank you very much for your careful review, our words are not very accurate here, according to your suggestion, we have modified “absorbed” into “stored” in line 154.

  1. L. 378. What a value of correlation?

Response:Thank you very much for your valuable suggestion. I’m sorry that the literature we cited only reported the trend of correlation changes but did not report the correlation value. From the original text, we can see that the trend is very significant. So unfortunately, I can’t provide a specific correlation coefficient value for you. Moreover, the correlation coefficient value is only a rough statistical parameter, I do not think it is meaningful to provide a specific value. Therefore, I sincerely hope that you will agree with my decision.

With best regards, Reviewer. Please keep going your research in soil erosion simulation, probably study with different slopes angle will have a interest for you. Good luck in future research!

Response:Thank you very much for your help in revising my paper. We will consider your suggestions carefully. In fact, we are preparing to do as you say in the future. Best wishes for you in the future.

Reviewer 3 Report

Dear authors, I highly appreciate work you done during the revision of the manuscript based on the comments from the reviewers. I don’t have any further comments.

Best regards.

Author Response

Dear Editors and Reviewers:

Thank you very much for the valuable time you spent for my article: "Slope runoff process and regulation threshold under the dual effects of rainfall and vegetation in loess hilly and gully region" (sustainability-2229925). We will try our best to do better in the future. Best wishes for you.